# Quantitative mapping of keratin networks in 3D

**Reinhard Windoffer[1]\*, Nicole Schwarz[1], Sungjun Yoon[1], Teodora Piskova[1,2], Michael Scholkemper[3], Johannes Stegmaier[4], Andrea Bönsch[5], Jacopo Di Russo[1,2,6], Rudolf E Leube[1]\***

[1]Institute of Molecular and Cellular Anatomy, RWTH Aachen University, Aachen, Germany; [2]Interdisciplinary Centre for Clinical Research, RWTH Aachen University, Aachen, Germany; [3]Department of Computer Science, RWTH Aachen University, Aachen, Germany; [4]Institute of Imaging and Computer Vision, RWTH Aachen University, Aachen, Germany; [5]Visual Computing Institute, RWTH Aachen University, Aachen, Germany; [6]DWI – Leibniz-Institute for Interactive Materials, Aachen, Germany

**\*For correspondence:**
rwindoffer@ukaachen.de (RW);
rleube@ukaachen.de (REL)

**Competing interest:** The authors declare that no competing interests exist.

**Abstract** Mechanobiology requires precise quantitative information on processes taking place in specific 3D microenvironments. Connecting the abundance of microscopical, molecular, biochemical, and cell mechanical data with defined topologies has turned out to be extremely difficult. Establishing such structural and functional 3D maps needed for biophysical modeling is a particular challenge for the cytoskeleton, which consists of long and interwoven filamentous polymers coordinating subcellular processes and interactions of cells with their environment. To date, useful tools are available for the segmentation and modeling of actin filaments and microtubules but comprehensive tools for the mapping of intermediate filament organization are still lacking. In this work, we describe a workflow to model and examine the complete 3D arrangement of the keratin intermediate filament cytoskeleton in canine, murine, and human epithelial cells both, in vitro and in vivo. Numerical models are derived from confocal airyscan high-resolution 3D imaging of fluorescence-tagged keratin filaments. They are interrogated and annotated at different length scales using different modes of visualization including immersive virtual reality. In this way, information is provided on network organization at the subcellular level including mesh arrangement, density and isotropic configuration as well as details on filament morphology such as bundling, curvature, and orientation. We show that the comparison of these parameters helps to identify, in quantitative terms, similarities and differences of keratin network organization in epithelial cell types defining subcellular domains, notably basal, apical, lateral, and perinuclear systems. The described approach and the presented data are pivotal for generating mechanobiological models that can be experimentally tested.

## Editor's evaluation

In this body of work you have skillfully adapted and developed tools for the three-dimensional visualization and systematic analysis of the entire keratin filament network in three different types of cells. The resulting contribution is original, provides insight at both a methodological and biological level, and nicely complements and extends emerging information about the high resolution structure of intermediate filaments in situ (by cryoelectron tomography / Medalia lab, Switzerland). The manuscript is well-written, well-illustrated, and the authors are thorough in their recognition of previous studies of relevance to their own. We believe that this article will be foundational in the specialized field of intermediate filament biology and will have a significant impact in the broad field of cell biology.

## Introduction

Quantification of cellular properties is a central task of cell biology (*Howard, 2014*; *Liberali and Pelk-mans, 2012*; *Lidke, 2017*; *Lippincott-Schwartz, 2014*; *Piston, 2014*; *Roca-Cusachs et al., 2017*). It is necessary for scientific analysis, testing, and model building. The structural diversity of eukaryotic cells supports their complex functions and is essential to provide local microenvironments, which facilitate and enhance the probability of highly specialized molecular interactions and reactions. The cytoskeleton is a key determinant of the necessary spatial organization. It consists of filamentous polymers, which assemble into microtubules, actin filaments, and intermediate filaments. The resulting networks are interlinked and connected to intra- and extracellular targets, including cytoplasmic organelles and specialized plasma membrane sites. In this way, they act as structural and mechanical connectors between extra- and intracellular forces. The participating structures and polypeptides have been referred to as the mechanobiome (*Kothari et al., 2019*; *Parajón et al., 2021*; *Wang et al., 2014*), which defines the mechanical setting of a cell. The properties of several mechanobiome components have already been quantified including their turnover and mechanical features at different levels. Comprehensive quantitative information on entire cytoskeletal networks, however, is missing for most components and, consequently, only little is known about the distribution, local composition, mesh size, orientation, and mechanical properties of cytoskeletal networks at the subcellular level. Therefore, there is an urgent need to describe cytoskeletal network architectures in quantitative terms at subcellular resolution. To this end, we present a novel approach to create the complete digital representation of keratin intermediate filament networks in three dimensions (3D), providing maps and quantitative measurements of the entire keratin network in cultured cells and in cells within their native tissue context.

The intermediate filament cytoskeleton consists of polypeptides, which are expressed in a tissue- and context-specific manner. The largest group of intermediate filaments, the keratin intermediate filaments, are the main cytoplasmic networks in epithelial tissues. They confer, together with their associated desmosomal cell-cell and hemidesmosomal cell-extracellular matrix adhesions, mechanical resilience (*Hatzfeld et al., 2017*; *Te Molder et al., 2021*). The human genome contains 54 keratin genes, which are expressed in epithelia and epidermal appendages in a cell type-specific pattern (*Jacob et al., 2018*). Keratin filaments are highly flexible and extensible (*Block et al., 2015*). They form hollow tubes with remarkable variability in local diameter and ultrastructure (*Weber et al., 2021*). Keratin filaments generate branched bundles of different thickness forming anisotropic networks (*Beil et al., 2005*; *Martin et al., 2016*). The 3D organization of keratin networks differs profoundly between different epithelia: for example, the simple epithelial cells lining the intestine contain a well-defined lateral and apical network, keratinocytes contain a dense 3D network encompassing the entire cytoplasm, and hepatocytes have a prominent submembraneous network (*Schwarz et al., 2015*; *Strnad et al., 2008*). Network organization is furthermore affected by mechanical stress (*Beriault et al., 2012*; *Felder et al., 2008*; *Fois et al., 2013*; *Karsch et al., 2020*; *Lutz et al., 2020*; *Quinlan et al., 2017*) and differentiation-status (*Coch and Leube, 2016*; *Iwatsuki and Suda, 2010*; *Leube et al., 2017*; *Quinlan et al., 2017*; *Tateishi et al., 2017*). In addition, individual proteins such as Ndel1 and plectin have been shown to regulate keratin network organization (*Kim et al., 2021*; *Moch et al., 2016*; *Osmanagic-Myers et al., 2006*). Knowledge of the 3D keratin network organization in single epithelial cells is therefore important to understand and characterize the particular functional status of a given cell within its complex tissue context under different conditions.

Segmentation of cellular elements is a key task in biological image analysis. Its application to specific structures, however, can be quite difficult and requires appropriate adjustment. Networks of curvilinear objects such as keratin networks are ubiquitous ranging from cytoskeletal filaments at the micro- to nanoscale to blood vessels or neuronal networks at the tissue and organ level (*Özdemir and Reski, 2021*). Filament detection systems have been developed for vascular (*Imran et al., 2019*; *Moccia et al., 2018*; *Samuel and Veeramalai, 2020*) and neuronal networks (*Acciai et al., 2016*; *Magliaro et al., 2019*; *Quan et al., 2016*). But these tools have not been optimized for subcellular structures and are specialized to detect branched, tree-like structures. The fishnet-like organization of keratin networks poses novel challenges, which are further confounded by the variably spaced branching points connecting filaments of different thickness and curvature.

A wide range of specialized tools are available for segmentation of cytoskeletal actin (*Alioscha-Perez et al., 2016*; *Nowak et al., 2020*), which also work in 3D (*Haspinger et al., 2021*). Further tools

have been developed for the segmentation of microtubules (*Faulkner et al., 2017*) and their prokaryotic homologue FtsZ (*Liu et al., 2019*; *Özdemir and Reski, 2021*). Recently, a tool was described that is suitable for segmentation of cytoskeletal networks including keratin networks using data obtained by electron microscopy and super-resolution fluorescence microscopy (*Flormann et al., 2021*). This tool, however, lacks 3D capability and does not capture information on filament thickness. Another segmentation tool for filamentous networks in electronmicroscopic images was also recently reported (*Dimchev et al., 2021*). Until now, however, no tools are available that segment 3D fluorescence image data of keratin filament networks and transform them into a numerical graph model.

To segment keratin filament networks and generate data structures with defined nodes and internodal regions ( = segments), we used the open-source program TSOAX. TSOAX has been applied for 3D segmentation of actin networks using fluorescence images of yeast cells (*Xu et al., 2019*). It is an implementation of SOAX (*Xu et al., 2015*; *Xu et al., 2011*), which was developed from the Stretching Open Active Contours (SOAC) algorithm (*Li et al., 2010*; *Li et al., 2009*). SOACs are open curves, which start at segment tips and delineate the filament centers. We now show that TSOAX can be used to reliably segment keratin networks in entire cells. TSOAX also provides density information of the identified keratin filaments. However, the association of nodes and segments is not detected satisfactorily as the generated TSOAX snakes are not separated consistently at filament nodes and therefore hinder statistical analysis. We therefore developed additional tools (KerNet) for optimal network representation of a node-segment structure. The combination of the established segmentation algorithms (TSOAX) and the additional KerNet analysis tools helped to create accurate numerical representations of keratin network topology.

The developed image analysis tools were applied to confocal airyscan high-resolution fluorescence image stacks of keratin networks in single cells either in cultured epithelial monolayers or within their native tissue context. We show that the image analysis tools provide cell-type-specific information on filament mesh characteristics as well as information on filament thickness, directionality, and connectivity in 3D. Cinematic rendering and immersive visualization make analysis of network heterogeneity accessible at a hitherto unknown quality and precision of 3D reconstruction.

## Results

### Recording of the keratin intermediate filament network in 3D

In a first step, we wanted to generate high-resolution fluorescence data that would be suitable for image analysis of single filaments and filament bundles and would represent keratin filament networks in different epithelial cell types. We selected three non-carcinogenic epithelial cell types: (*i*) Canine kidney epithelial MDCK cells producing YFP-tagged keratin 8 (subclone H9) were chosen as representatives of polarized simple epithelial cells. MDCK cells form a one-layered coherent epithelium with the cuboidal to cylindrical cells attached to each other by highly organized junctional complexes. Intracellularly, the keratin filaments have a complex arrangement consisting of a dense subapical network, perinuclear and radial keratin filaments, and interdesmosomal filaments below the plasma membrane (*Quinlan et al., 2017*). Immunoblot analysis of total cell lysates revealed comparable amounts of YFP-tagged and endogenous keratin 8 (*Figure 1—figure supplement 1*). (*ii*) Spontaneously immortalized human epidermal keratinocytes expressing YFP-tagged keratin 5 (HaCaT B9) were chosen as representatives of squamous epithelial cells (*Boukamp et al., 1988*; *Moch et al., 2013*). They contain pancytoplasmic keratin filament networks whose 3D organization is only partially elucidated (*Moch et al., 2020*). (*iii*) To examine the 3D organization of the keratin intermediate filament network in its native tissue context, we made use of the recently described knock-in mouse line expressing only YFP-tagged keratin 8 but no wild-type keratin 8. Retinal pigment epithelium cells (RPE) were selected for the analysis because of their well-delineated cytoplasmic keratin filament network (see below). High-resolution airyscan confocal image stacks were prepared from each cell type.

### Transformation of 3D fluorescence recordings of keratin filaments into digital network models

The recorded image stacks were transformed into Euclidean 3D maps using TSOAX and the KerNet software (*Figures 1 and 2*; *Figure 1—figure supplements 1 and 2*). In these maps, curved segments that are connected by nodes describe the entire keratin networks of single cells. The segments

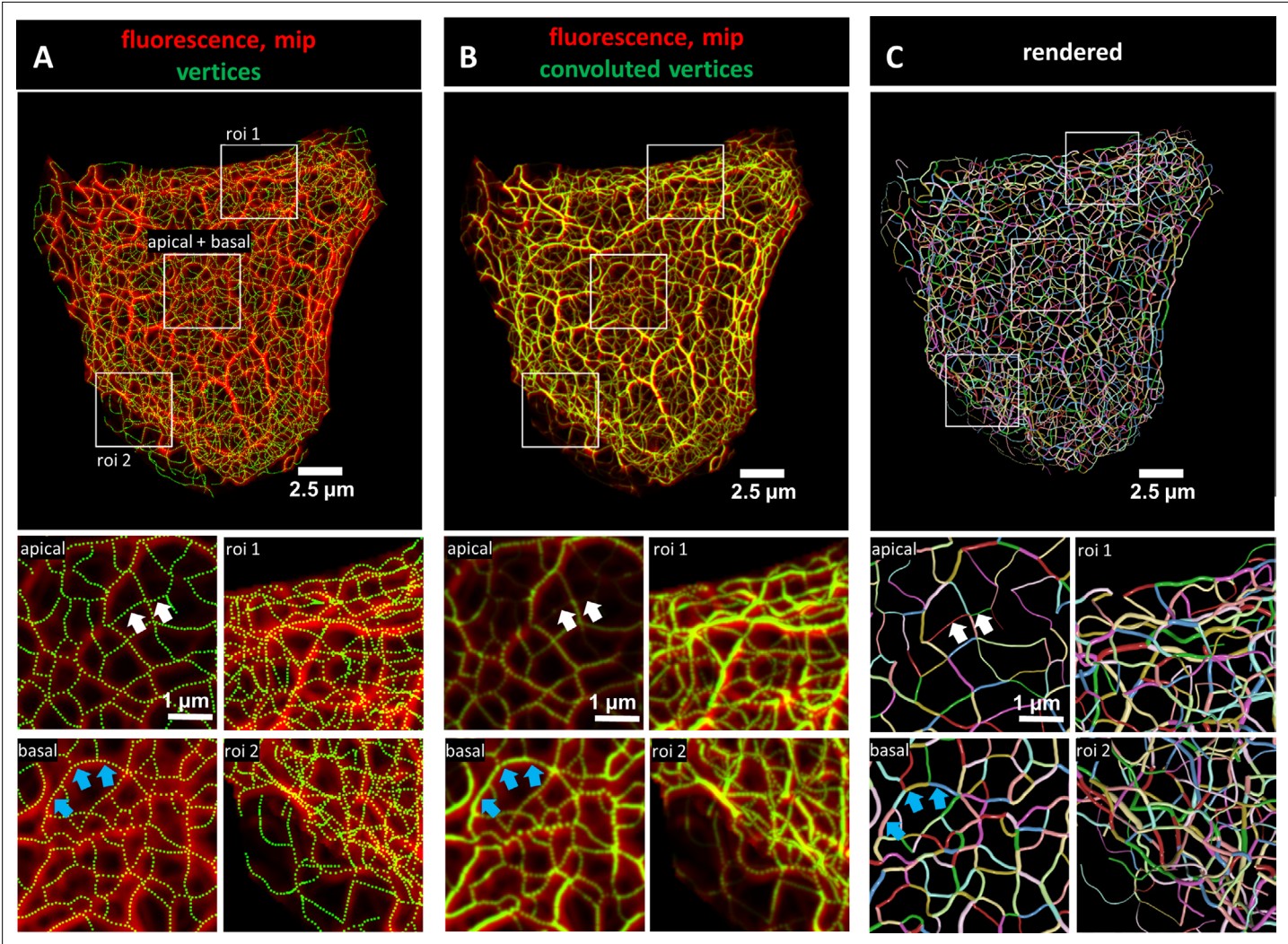

**Figure 1.** Evaluation of 3D keratin filament network segmentation. The pictures show three different numerical representations of the same fluorescence data set detecting the keratin filament network in an MDCK cell expressing fluorescently labeled keratin 8. The cell is part of a confluent monolayer and the fluorescence of the single cell shown was manually excised from an image stack consisting of 32 slices (voxel size: xy = 66 nm, z = 182 nm). This cropped fluorescence was used to calculate the vertex positions and intensity of all keratin filaments. (**A**) The top picture shows an overlay of the recorded fluorescence in red as a maximum intensity projection (mip) and the corresponding calculated xy positions of vertices of all segments as green dots. The enlargements below (corresponding regions demarcated by boxes) show maximum intensity projections of focal planes above and below the nucleus (apical and basal) at left and two regions of interest (roi 1, roi 2) depicting maximum intensity projections of all focal planes in the cell periphery. (**B**) Presents an overlay of the keratin fluorescence and corresponding vertices that were plotted in 3D with a diameter relative to their brightness and were subsequently subjected to 3D Gaussian blurring to simulate the microscope's blur. The enlargements correspond to those shown at left in (**A**) presenting the same regions of interest and focal planes. The top panel of *Video 1* presents an animated comparison of the fluorescence recording, segmentation and overlay in each focal plane. (**C**) Depicts a cinematic rendering of segments. Each calculated segment is rendered in 3D with a thickness corresponding to the brightness of the original filament. The color-coding helps to identify and distinguish individual segments. The regions of interest correspond to the same single planes as those shown in (**A**). White arrows demarcate thin filaments, blue arrows thick filaments.

The online version of this article includes the following figure supplement(s) for figure 1:

**Figure supplement 1.** Immunoblot of a total lysate of a transgenic MDCK cell clone detecting endogenous wild-type keratin 8 (Krt8) and eYFP-tagged keratin 8 (Krt8-eYFP).

**Figure supplement 2.** Exemplary representation of major steps in the workflow.

**Figure supplement 3.** Validation of segmentation.

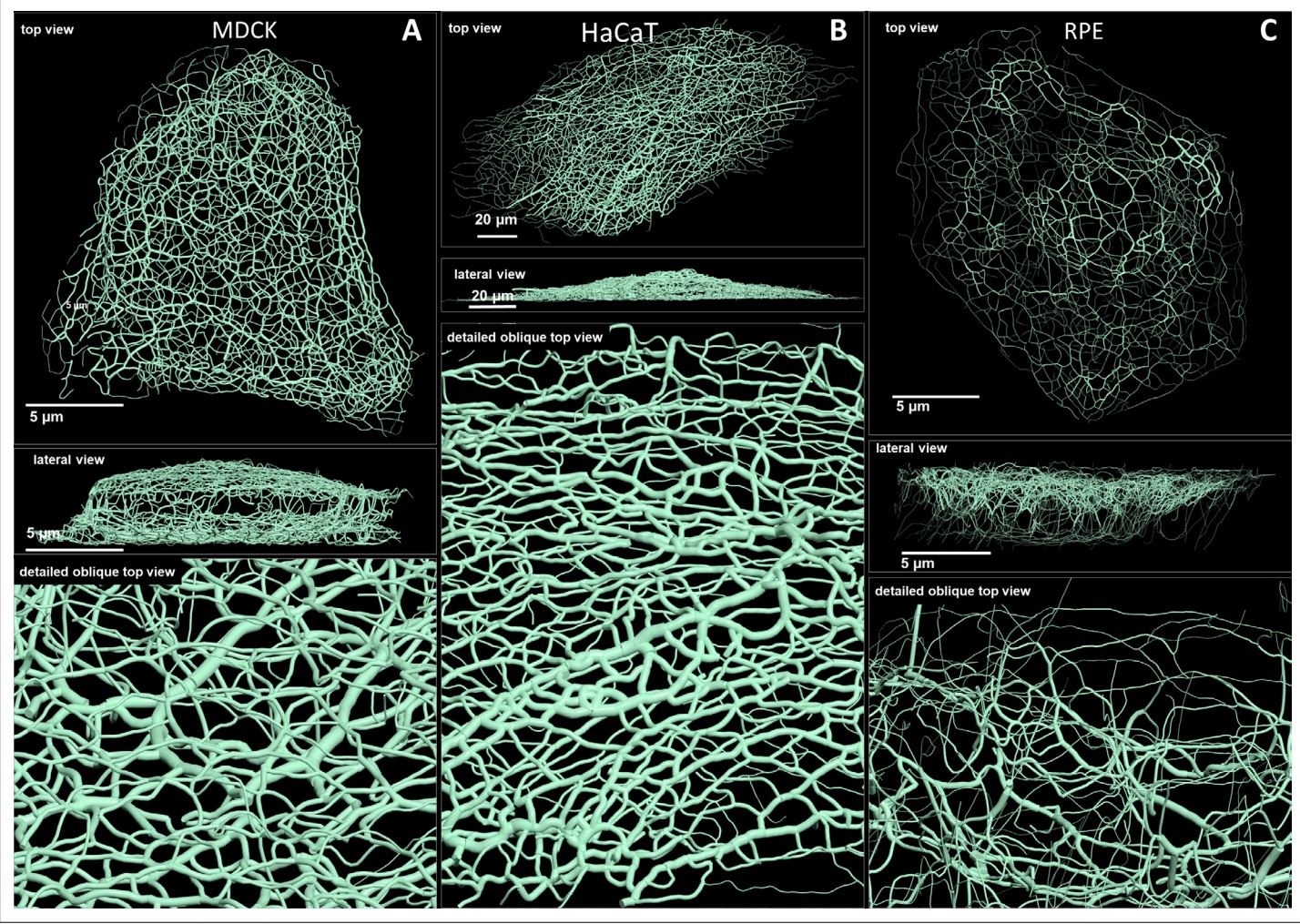

**Figure 2.** Comparison of 3D cinematic rendering of keratin filaments in three different epithelial cell types. The position and intensity of fluorescence recordings were transferred into a numerical 3D-representation of individual segments. The thickness of the segments corresponds to the fluorescence brightness in the original data. (**A**) Shows cinematic rendering of keratin filaments of an MDCK cell expressing fluorescently labeled keratin 8. *Video 2* presents the 3D-animation of the cell. (**B**) Depicts the cinematic rendering of keratin filaments in a HaCaT keratinocyte expressing fluorescently labeled keratin 5 (animation in *Video 3*). (**C**) Illustrates cinematic rendering of keratin filaments in a single RPE cell of a homozygous keratin 8-YFP knock-in mouse (animation in *Video 4*).

The online version of this article includes the following figure supplement(s) for figure 2:

**Figure supplement 1.** Visual inspection of the keratin network of a single cell in a virtual reality environment.

represent keratin filament bundles of different thickness and length. These segments are defined as 3D polygonal chains of vertices, each of which has a specific xyz position and brightness. The first and last vertices of a segment are defined as nodes. Nodes are either shared by adjacent connecting segments within a filament or are positioned at the end of a filament in case of loose ends. Note that keratin filaments consist of multiple consecutive segments. To validate the quality of the numerical transformation, all vertices were plotted onto the maximum intensity projection of the fluorescence image stacks. *Figure 1A* shows the maximum intensity projection of the fluorescence in an MDCK cell as an example highlighting selected regions of interest at higher magnification. Separating the regions above and below the nucleus (apical and basal, respectively) furthermore allowed to distinguish the different keratin network structures in these subcellular domains. Visual inspection showed a high degree of overlap between the recorded fluorescence and the derived numerical positions of vertices (dotted lines) at all subcellular locations.

To obtain more detailed comparisons of the calculated vertices with the fluorescence data, vertices were first plotted in 3D and their brightness was encoded as filament diameter. The 3D plots were

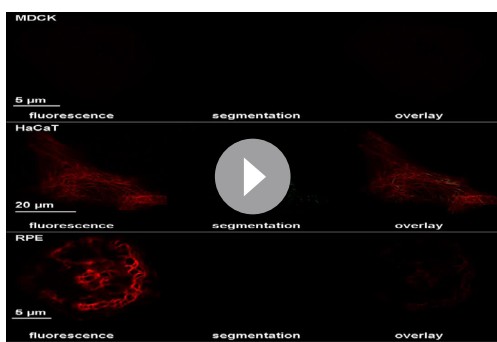

**Video 1.** Validation of keratin filament segmentation in an MDCK, HaCat and RPE cell. The animation shows the original fluorescence image stacks (left), the segmented data including the associated brightness (middle) and the overlays of both (right).
https://elifesciences.org/articles/75894/figures#video1

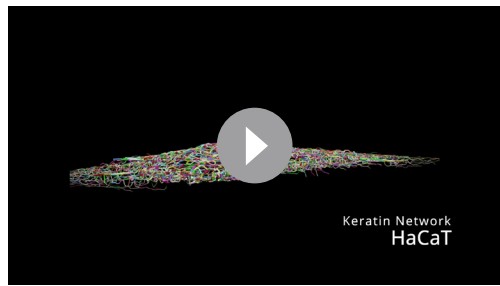

**Video 3.** Cinematic rendering of the numerical reconstructed keratin network of a HaCaT cell growing in a confluent monolayer. The tubes represent keratin bundles with different thickness. At the beginning of the video, the segments are randomly color-coded.
https://elifesciences.org/articles/75894/figures#video3

then convolved by Gaussian blurring to simulate the microscope's point spread function. *Figure 1B* shows the very good match of the convolved vertices in terms of position and brightness in comparison to the maximum intensity projection of the fluorescence images. To assess the overall reliability of the digital transformation, the validation steps were applied to all cells selected for the study (*Figure 1—figure supplement 3*). Additional slice-by-slice comparisons are presented exemplarily in *Video 1* further demonstrating that the recorded and derived numerical 3D representations of the keratin filament network are in agreement.

For close inspection of the keratin filament network in its subcellular complexity, we used cinematic rendering. The rendering displays the network as a true 3D model including the brightness of filaments which are shown as tubes with different degrees of thickness thus providing information on the segments in a 3D-viewer. To distinguish individual segments, each segment was randomly color-coded (*Figure 1C*).

## Comparison of 3D cinematic keratin filament renderings in MDCK, HaCaT, and RPE cells

Renderings of keratin filament networks in MDCK, HaCaT, and RPE cells are exemplarily shown from different views in *Figure 2* and *Videos 2–4*. Interactive 3D renderings of all cells are available at kernet. rwth-aachen.de. Furthermore, immersive visualization was used to provide unrivaled impressions of the spatial arrangements of the keratin network in stereoscopic 3D (*Figure 2—figure supplement 1*). The multidimensional visualizations revealed hallmark features of the different keratin filament networks.

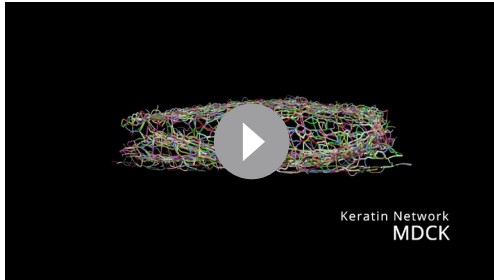

**Video 2.** Cinematic rendering of the reconstructed keratin filament network of an MDCK cell growing in a confluent monolayer. The tubes represent keratin bundles with different thickness. At the beginning of the video, the segments are randomly color-coded.
https://elifesciences.org/articles/75894/figures#video2

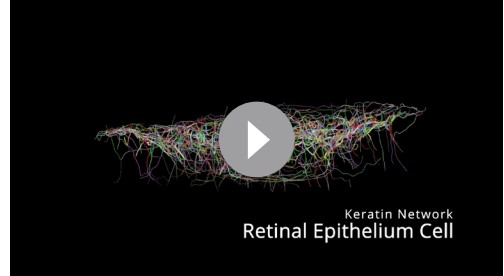

**Video 4.** Cinematic rendering of the numerical reconstructed keratin network of an RPE cell. The tubes represent keratin bundles with different thickness. At the beginning of the video, the segments are randomly color-coded.
https://elifesciences.org/articles/75894/figures#video4

Thus, an apical and a basal keratin filament network domain can be distinguished in MDCK cells (*Video 2*). The major difference between both domains are the thicker keratin bundles in the basal domain. Each subcellular network is characterized by a regular mesh pattern. The lateral keratin filament network is more heterogenous and dominated by filaments running between apical and basal. Circumferential subcortical keratin filaments are mainly restricted to the apical and basal network domains. In the cell interior, keratin filaments form a network around the nucleus. It is best developed on the basal side with highly bundled filaments, many of which are rather straight.

HaCaT cells are extremly flat and do not show a clearly seperated apical and basal network and lack a distinct lateral network (*Video 3*). A prominent feature of their keratin filament network are long filament bundles that often run in parallel. They are absent above and below the nucleus but enclose the nucleus laterally.

The keratin network is less dense in RPE cells than in the other two cell types (*Video 4*). It is prominent in the cytoplasmatic apical domain, which faces the photoreceptor cells in the retina. It is also very well developed around the nucleus. In contrast to MDCK cells, the perinuclear network contains thicker filaments at the apical than the basal side. The basal cytoplasmatic domain is mostly devoid of keratin filaments. Some filaments run through the cytoplasm from basal to apical.

In conclusion, rendering of the keratin filament networks in three cell types revealed very different arrangements. The numerical data representation now allows detailed further analyses of distinct features. Especially with the help of Virtual Reality (VR), classification of network domains becomes feasable.

## Comparison of keratin filament brightness distribution patterns

The numerical network representations were used to quantify keratin filament properties. A prominent property, which was already graphically encoded in the 3D-models, is filament brightness as a measure of filament thickness. The assumption is that filament thickness is a reliable indicator of filament bundling which can now be easily quantified throughout entire cells. We will therefore use the terms brightness and thickness interchangeably. The histograms in *Figure 3A–C* depict the spread of filament segment brightness in MDCK, HaCaT, and RPE cells. The quantifications show that filament segments with a medium brightness dominated in MDCK cells with a near Gaussian distribution, whereas thin filaments were prevalent in HaCaT and RPE cells. Thin filaments were most abundant in HaCaT cells.

Maps of filament brightness are shown in *Figure 3D–F* for selected examples of each cell type at low and high magnification. In each case, thin filaments were numerous in the cell periphery. This may be due to the increased de novo single filament formation in these regions (*Windoffer et al., 2004*; *Windoffer et al., 2006*, *Moch et al., 2013*). Some areas, such as those shown at the right margin of the MDCK cell in *Figure 3D*, however, presented rather thick keratin filament bundles, which may represent matured interdesmosomal keratin filaments (*Quinlan et al., 2017*). The most heterogeneous network in terms of filament thickness was encountered in MDCK cells. The animation in *Video 2* clearly shows that filaments in the basal network of the cell are thicker than that in the apical network. In HaCaT cells, enrichment of thin filaments was visible around the nucleus (*Figure 3E*) whereas thicker filaments were detectable in the cytoplasm, many of which appeared to be arranged in parallel (see also *Video 4*). In contrast, thick filament bundles surrounded the nucleus of RPE cells (*Figure 3F*; *Video 4*). Furthermore, thick bundles were more frequent above the nucleus, that is at the part of the cell facing the photoreceptors. Together, the analyses of the digitalized networks revealed major differences in the extent and distribution of keratin filament bundling suggesting that the keratin network is organized to withstand and cope with different types of mechanical stress.

## Number and length of keratin filament segments and overall keratin filament length in single cells

The number and length of individual keratin filament segments are key measurements of network organization. Significantly more segments were detected in HaCaT cells than in the other cell types (*Figure 4A*). The average segment length was shortest in MDCK cells (0.80 ± 0.04 μm), longest in RPE cells (0.88 ± 0.05 μm) and intermediate in HaCaT cells (0.83 ± 0.13 μm) (*Figure 4B*). The combined length of all segments per cell (*Figure 4C*) was similar in MDCK (1.63 ± 0.41 mm) and RPE cells (1.57 ± 0.48 mm) but was more than twice longer in HaCaT cells (4.40 ± 1.53 mm). To see whether this

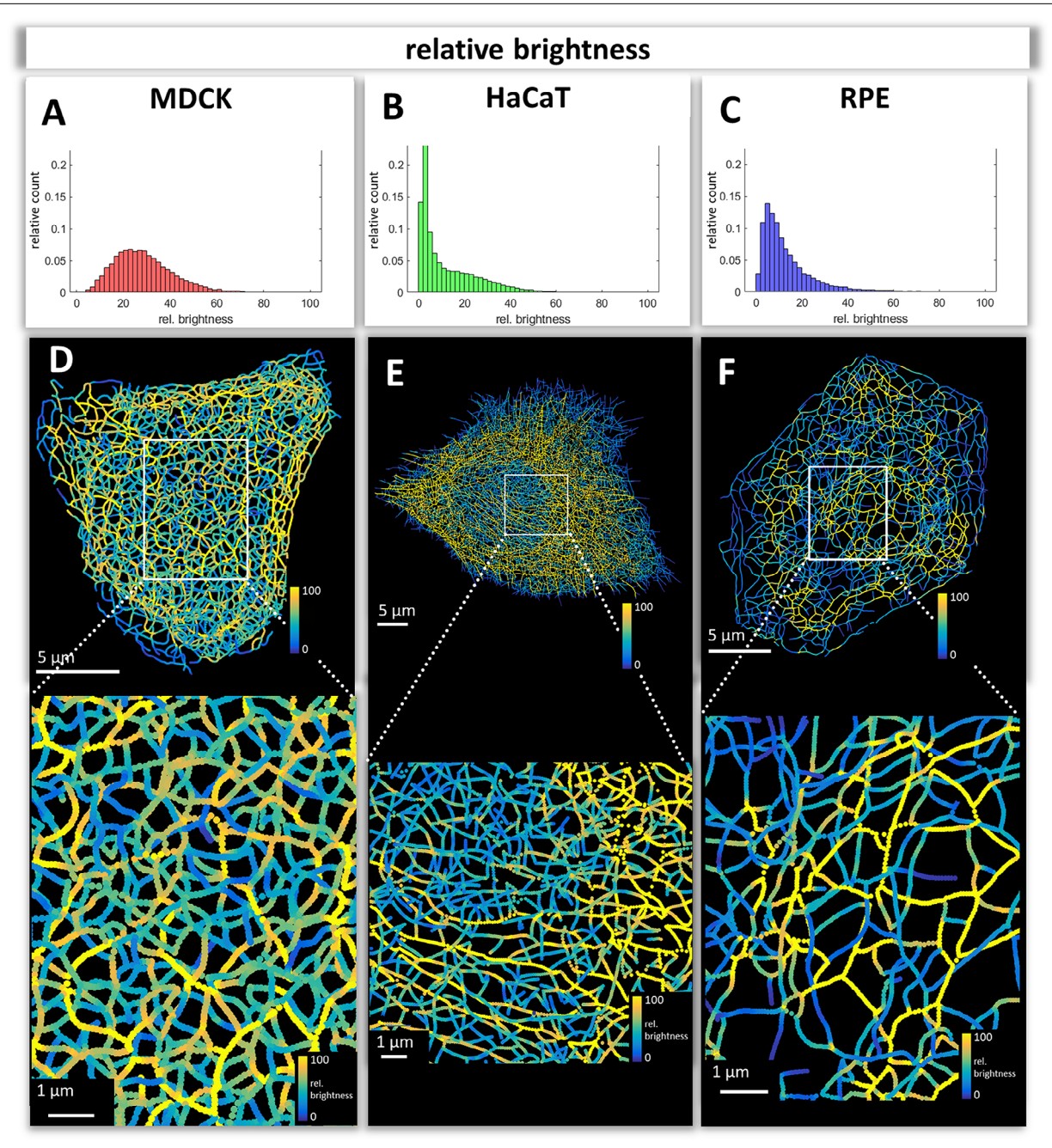

**Figure 3.** Mapping of keratin filament brightness in MDCK, HaCaT, and RPE cells. (**A–C**) The histograms display the distribution of relative segment brightness for each cell type. The maximum brightness was normalized to 100 in all cases. For each segment, the mean brightness was calculated and the determined brightness levels of all segments in 10 cells were pooled for each diagram. (**D–F**) The maps depict the segment brightness in a representative MDCK (left), HaCaT (middle) and RPE (right) cell (top panel; enlarged boxed area in bottom panel). The color code represents relative brightness.

difference was due to different cell dimensions, the cell volumes were determined for all three cell types. Significant differences could be detected, i.e., 4.65 ± 1.13 µm³ for MDCK, 9.16 ± 6.10 µm³ for HaCaT and 2.73 ± 0.95 µm³ for RPE (*Figure 4—figure supplement 1A*). Taking the different cell volumes into account, filament density, that is filament length per volume, was comparable in HaCaT (604.82 ± 304.20 µm/µm³) and RPE cells (585.60 ± 63.97 µm/µm³), while it was considerably lower in MDCK cells (355.24 ± 56.49 µm/µm³) (*Figure 4D*). Given that the human body has ~0.14 x 10¹² epidermal cells (*Bianconi et al., 2013*), the entire length of epidermal keratin filament bundles in

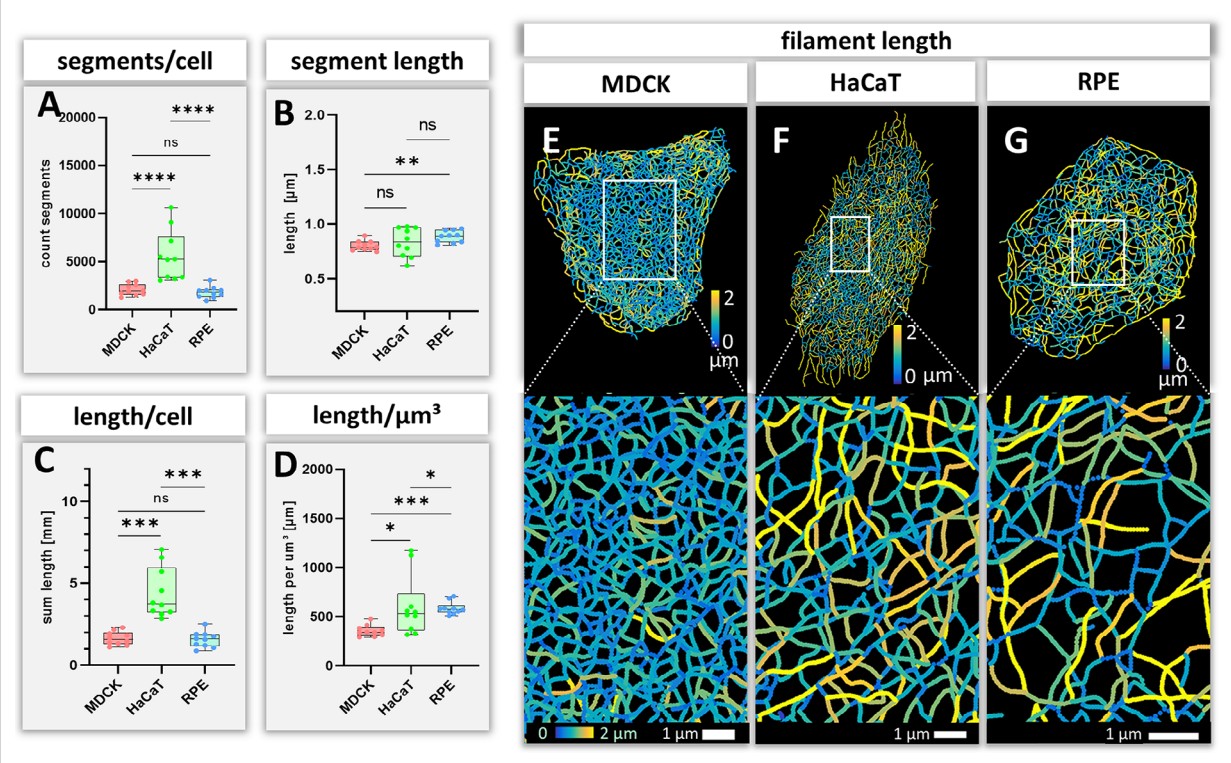

**Figure 4.** Measurements of keratin filaments in MDCK, HaCaT and RPE cells. (**A–D**) The whisker box plots depict the number of segments per cell (**A**), the mean segment length (**B**), the entire length of the keratin filament network within a single cell (**C**) and the filament length per μm³ (**D**). Ten cells were analyzed for each cell type. (**E–G**) show color-coded keratin filament segment lengths for each cell type. The top panel presents the results for entire cells, the bottom panel the results for the boxed areas at higher magnification.

The online version of this article includes the following figure supplement(s) for figure 4:

**Figure supplement 1.** Boxplots of cell volumes estimated from fluorescence image stacks (**A**) and calculated virtual persistence length of segments (**B**) in MDCK, HaCaT and RPE cells.

an individual should be ~600,000 km. Assuming a bundling rate of 19 filaments per filament bundle (*Hémonnot et al., 2016*) there are ~$1.14 \times 10^7$ km of single keratin filaments in the epidermis of a human body.

## Estimation of cellular keratin content

It is now possible to estimate the amount of keratin from our observations and to compare it to the previously estimated 8.07 ± 1.88 pg keratin in single basal keratinocytes of the murine epidermis (*Feng et al., 2013*). To this end, we take the experimentally determined mass per length (MPL) of keratin filaments of 25 kDa/nm (range: 19–30 kDa/nm; *Herrmann et al., 1999*) and a bundling factor (Bf) of 19 (*Hémonnot et al., 2016*) into account. As shown in the previous section, the total length of keratin filament bundles ($L_{tot}$) in HaCaT cells is ~4.4 mm. By using the following formula (($MPL * L_{tot} * Bf) / N_a$) [$N_a$ = Avogadro constant], the amount of keratin per cell is 3.47 pg.

An alternative estimate can be obtained by considering the dimensions and mass of the tetrameric subunit. A keratin tetramer has a length of 44 nm (*Parry et al., 2007*) and consists primarily of two keratin 5 and two keratin 14 polypeptides with a combined molecular weight of 228 165 Da in keratinocytes. Thus, 22.72 tetramers make up a 1 μm long protofilament. It has been previously shown that ~6 protofilaments are contained within a 10 nm keratin filament (*Weber et al., 2021*). Thus, a 1 μm-long keratin filament has a MW of 1.24E + 08 Da. Assuming a bundling factor of 19 and a total keratin filament bundle length of ~4.4 mm in HaCaT keratinocytes, the total MW of keratin/cell is 2,60E+12 Da corresponding to a mass of ~4.3 pg/cell.

Performing the same type of calculation we deduced a keratin content of 1.3 pg or 1.4 pg per MDCK cell and 1.2 pg or 1.4 pg for single RPE cells.

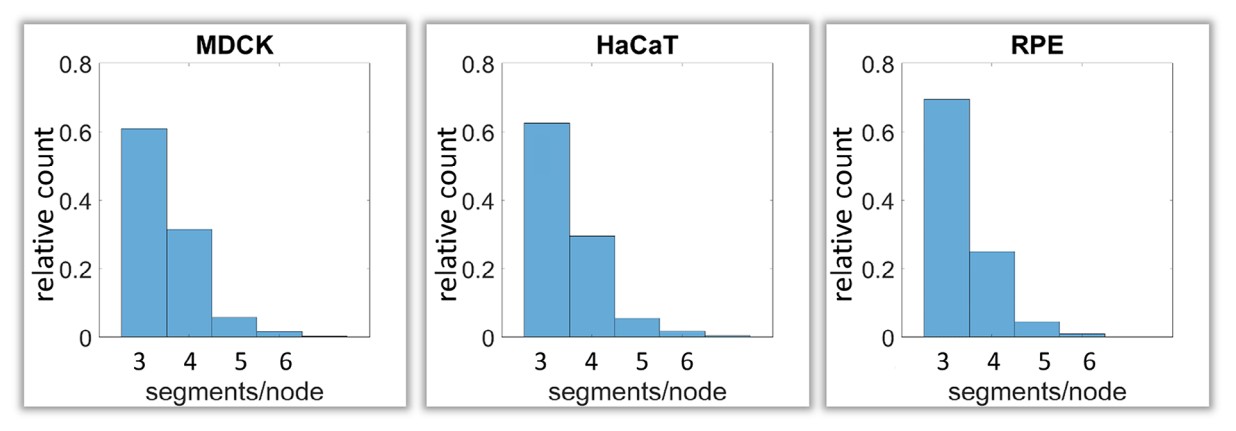

**Figure 5.** Calculation of keratin filament node types based on segment/node ratios in MDCK, HaCaT and RPE cells. A segment/node ratio of 3 refers to Y-shaped branching, a ratio of 4 to X-shaped branchings. Higher values refer to star-like branchings. The histograms depict the relative distribution of node types.

## Branching and intersection of keratin filaments

Next, we analyzed branching and intersection of keratin filament bundles. Different types of branching could be envisioned depending on the number of segments per node. Three segments per node would define a Y-shaped branch, whereas four segments would define an X-shaped intersection of two filaments (*Figure 1—figure supplement 2C*). More than four branches would result in a star-like configuration. Our analyses show that the Y-shaped organization (three segments per node) was by far the most frequent configuration (~60%) followed by the X-shaped (four segment per node) arrangement (~30%) and rarely higher degrees of branching (~10%) (*Figure 5*).

## Bending of keratin filaments

Bending of keratin intermediate filaments is of interest as it may provide information on material properties of filaments and may indicate whether filaments are under longitudinal compression or tension. First, we calculated the distance/length ratio of segments (*Figure 6A*). The highest ratios were observed in MDCK (0.92 ± 0.09) and HaCaT (0.92 ± 0.08), which were significantly higher than in RPE cells (0.90 ± 0.10) indicating that the filaments in RPE cells are less straight. This interpretation was confirmed by curvature determination of segments (*Figure 6B*). The mean curvature of segments was significantly higher in RPE cells (3.30 ± 0.15 μm⁻¹) than in MDCK cells (2.54 ± 0.11 μm⁻¹) and HaCaT cells (2.97 ± 0.72 μm⁻¹). The curvature distribution in HaCaT cells was more widely spread than in the other two cell types. Maps of curvature distribution are shown from two different perspectives in *Figure 6C–E* revealing complex patterns. We did not see a correlation between filament bundle thickness and curvature (pairwise Pearson correlation coefficient in MDCK = –0.10, HaCaT = 0.08, RPE = 0.14). Finally, we determined the apparent persistence length of keratin filament segments being aware that the resulting values are difficult to interpret, since segments are not free to move but are connected to other segments within the network. *Figure 4—figure supplement 1B* shows that the calculated values (~2.6 μm) do not differ significantly between the three cell types. As expected, they are higher than the experimentally determined persistence length of single filaments (cf. *Block et al., 2015*).

## Keratin filament segment orientation

To calculate how evenly the segments are orientated in space, the azimuth ( = horizontal orientation) and the elevation ( = vertical orientation) of segments were calculated for each cell (*Figure 7—figure supplements 1 and 2*). The histograms in *Figure 7A–F* show examples of the azimuth and elevation distribution in single MDCK, HaCaT, and RPE cells. Comparing the determined values with the expected values for segments orientated evenly in all directions, shows that the azimuth distribution is quite even in MDCK and RPE cells but not in HaCat cells. All cells have an uneven distribution of vertical orientation, indicating that segments are preferentially oriented horizontally. Compilations

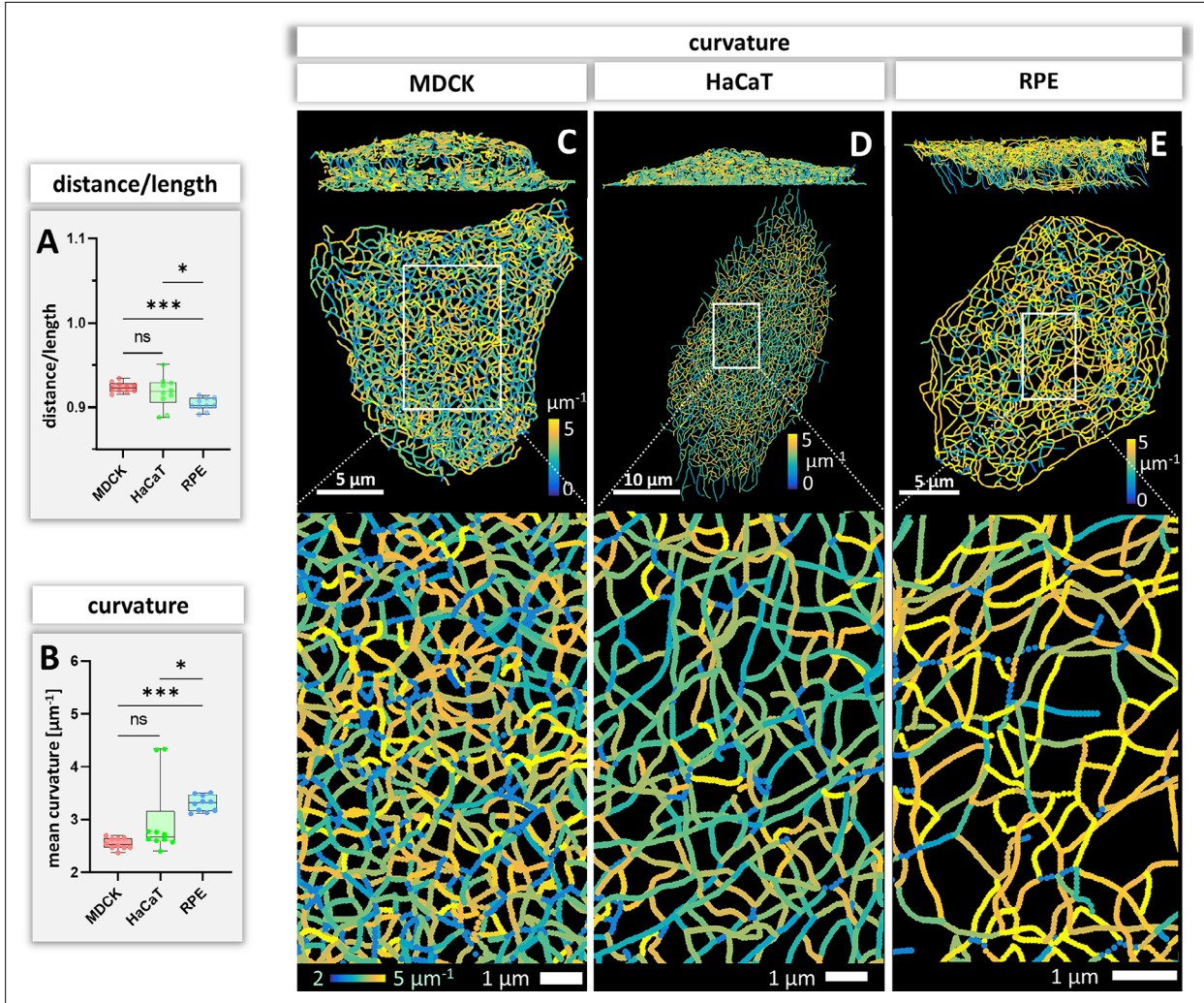

**Figure 6.** Measurements of keratin filament segment bending in MDCK, HaCaT, and RPE cells. (**A**) The whisker box plot shows the distance/length ratio of keratin segments. (**B**) The curvature of keratin filaments in the three cell types is shown as a whisker box plot. (**C–E**) Color-coded representation of the keratin network depicting the curvature of keratin filament segments in entire cells (top) and selected regions of interest (boxed areas) at higher magnification (bottom).

of the analyses for all cells (*Figure 7G and H*) confirmed the single-cell data and affirmed that the azimuth distribution in HaCaT is more oriented than in MDCK or RPE cells. Furthermore, the elevation orientation is more directed in MDCK and HaCaT cells than in RPE cells. Mapping the horizontal segment orientation in selected cells (*Figure 7I–L*) reveals a random pattern in MDCK and RPE cells and highlights parallel-arranged filaments in HaCaT cells.

To obtain insight about the relationship between segment directionality and cell shape, the orientation of the segments was analyzed with respect to the cell center. To this end, all segments were translocated to the cell center and their spatial distribution was plotted in radial 2D histograms. As expected, MDCK and RPE cells showed a homogenous distribution of horizontal segment orientation in the x/y plane while HaCaT cells were more polarized (*Figure 8A–C*). In the vertical direction (y/z plane), horizontally oriented segments predominate in MDCK and HaCaT cells while segments are more uniformly oriented in RPE cells (*Figure 8D–F*).

To compare the 3D-isotropy of entire keratin filament networks, a sum vector of all segments was constructed for each cell. To this end, the orientation vectors of segments were translocated to the cell center as described above and mapped with the length of each segment. *Figure 8G–I* shows the resulting sum vectors in each plane together with corresponding views of the segmented keratin filament networks. The sum vectors in MDCK and RPE cells are relatively small, indicating an isotropic

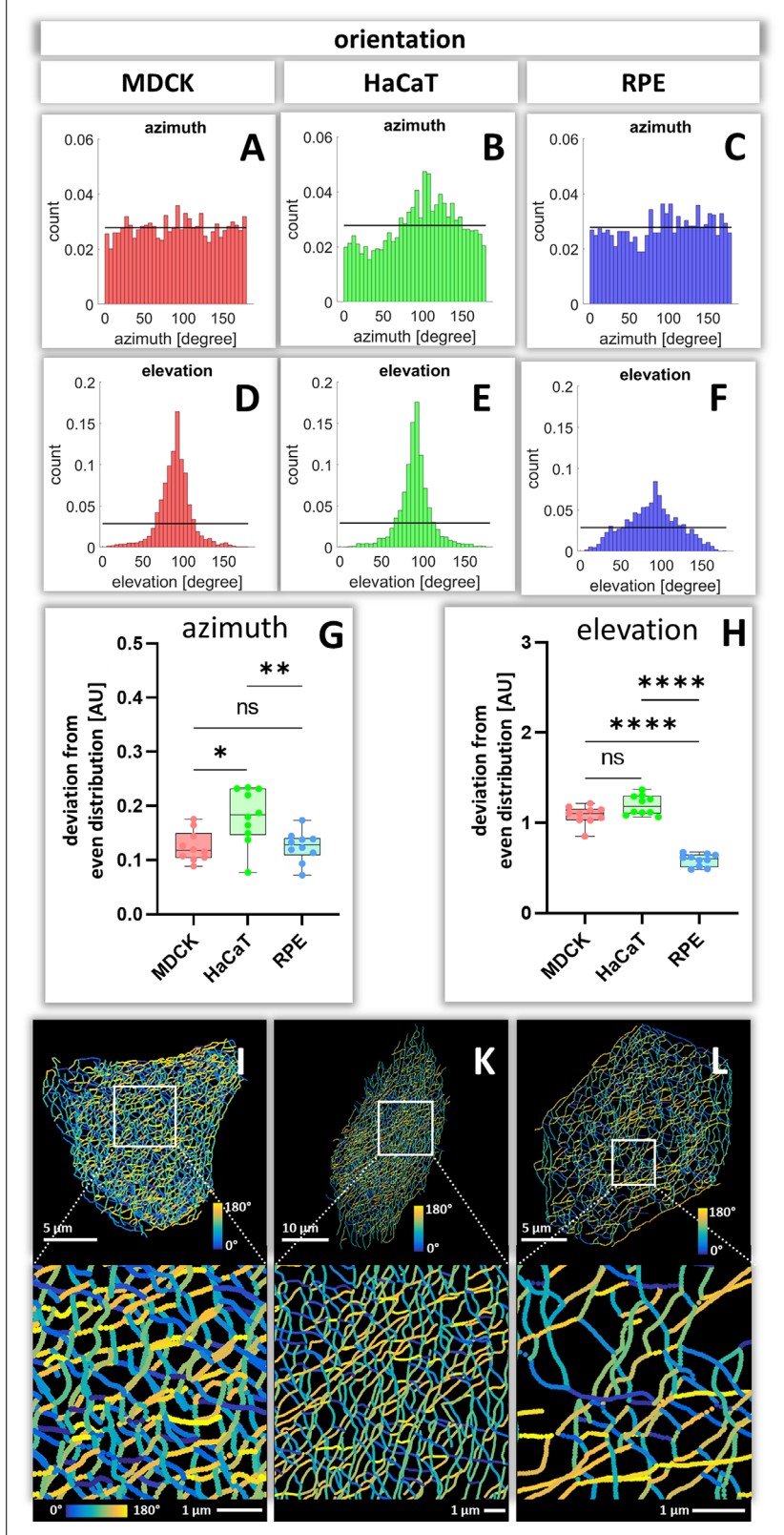

**Figure 7.** Orientation of keratin filament segments in MDCK, HaCaT, and RPE cells. (**A–C**) The histograms show the normalized distribution of azimuth ( = horizontal orientation) of keratin filament segments in single cells. Since the start and end points are interchangeable, the histogram is restricted to 180 degrees. The horizontal line represents the expected histogram levels, if all angles were equally distributed. (**D–F**) The histograms show the elevation ( =

*Figure 7 continued on next page*

*Figure 7 continued*

vertical orientation) of keratin filament segments. The horizontal line demarcates the expected histogram values, if elevation of keratin filament segments would be uniform. (**G, H**) Whisker box plots depicting the deviation of segment orientations from uniform unbiased distributions. The differences in azimuth (**G**) and elevation (**H**) were calculated for 10 MDCK, 10 HaCaT and 10 RPE cells. (**I–L**) The color-coded maps depict the azimuth of segments in entire cells (top; MDCK, left; HaCaT, middle; RPE, right) and selected regions of interest (boxed areas; bottom).

The online version of this article includes the following figure supplement(s) for figure 7:

**Figure supplement 1.** Schematic illustration of keratin filament segment (green) orientation in 3D space.

**Figure supplement 2.** Histograms of keratin filament orientation in MDCK, HaCaT, and RPE cells.

orientation of the respective keratin networks. In contrast, the sum vectors in HaCaT cells are much longer, indicating a higher degree of anisotropy of the keratin network (*Figure 8K*).

## Discussion

With this work we provide, for the first time, experimentally derived numerical 3D representations of entire keratin networks in eukaryotic cells. 3D fluorescence data were used to define segments that are connected by nodes. This enabled us to compare local properties of keratin intermediate filament networks such as filament thickness, curvature and orientation in different cell types in vitro and in vivo. It further facilitated comprehensive comparisons of entire networks by quantifying parameters such as entire filament length, degree of filament connectivity, and spatial network orientation.

A critical aspect of the work was evaluation of the quantification process. Since synthetic 3D data of cellular keratin networks were lacking, the quality of segmentation was evaluated by visual inspection of superimposed imaging and segmentation data demonstrating that the location and brightness of filaments were mapped with high precision and reliability. The subsequent allocation of nodes and segments was visually assessed in rendered 3D models (cinematic rendering) again revealing agreement of the assignments. Nevertheless, the numerical representation has its limitations in accuracy. Proper segment and node detection was particularly challenging in HaCaT cells because of their complex network geometry in combination with low filament bundling. Extreme bending at the single filament level with bending radii of 100 nm (*Weber et al., 2021*) are not detected in the present set-up due to the resolution limit of the airyscan. Mapping of keratin filaments at the cell edges was also difficult in some regions, mostly because of the manual assignment of cell borders in the confluent cell layers with tightly attached and partially overlapping neighboring cells. In cultured cells, mixing of fluorescent and non-fluorescent cells could overcome this problem. Segmentation quality may be further improved in the future by using other and more powerful super-resolution techniques than the airyscan technology used in the present work.

A major advance of the current report is the fabrication of numerical models of the keratin filament network in 3D. The rendered 3D models can be thoroughly inspected with the help of 3D display devices such as 3D screens, virtual reality-headsets (HMDs) or Cave Automatic Virtual Environments (CAVEs). The benefits of such immersive environments have been demonstrated (*Cruz-Neira et al., 1993*; *Laha et al., 2013*). Three factors thereby improve the domain scientists' spatial judgment and thus the task performance during the visual analysis: (i) an enlarged field of regard compared to standard desktop systems reduces clutter during data exploration, (ii) stereoscopy provides supporting depth cues, and (iii) head tracking provides enhanced motion parallax. Motion parallax is a monocular depth cue which allows observers to perceive the depth of the visualized cell from their own motion. The filaments close to the observers are perceived to be moving faster than filaments in the distance.

The derived digital network models furthermore allow comparisons between cells with similar or dissimilar differentiation status (*Figure 9*). As a proof of concept, we paradigmatically investigated three different epithelial cell types finding significant differences in filament bundling, segment length, filament density, filament curvature and filament orientation. These parameters are a crucial and first step to build mechanical models of keratin filament networks in different cell types and functional contexts. The following list shows potential implications of the extracted features for our understanding of keratin filament network properties and functions:

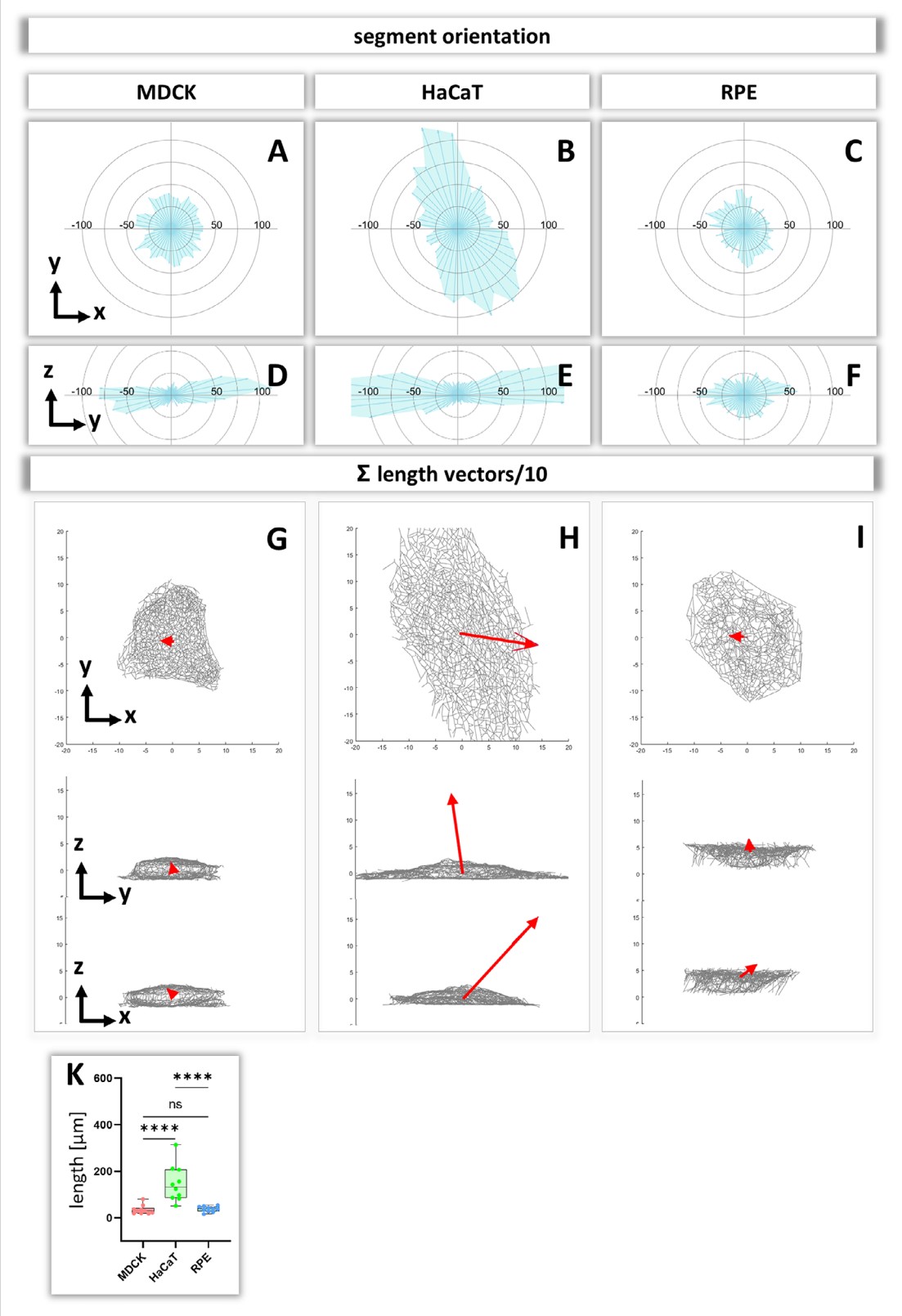

**Figure 8.** Analysis of segment orientation in MDCK, HaCaT, and RPE cells. (**A–F**) The blue polygons represent radial histograms of the orientation of keratin filament segments in relation to the cell center of single cells. Horizontal orientation (**A–C**) and the vertical orientation (**D–F**) are shown. Each keratin segment in each cell was moved to the cell center and vectors pointing in the same directions were added. Thus, the longer the blue lines are, the more vectors point in that direction. (**G–I**) The red arrows represent the sum vectors of the direction and length of all segments in each cell. (**K**) Whisker box plots depicting the length of the main vectors in all three cell types.

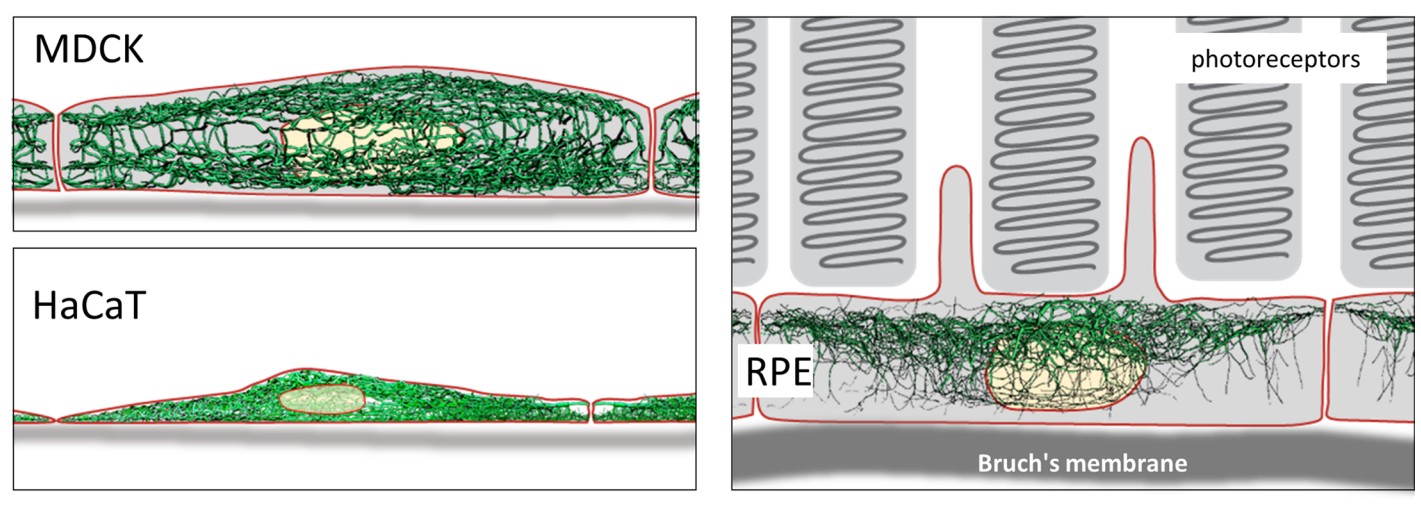

**Figure 9.** Schematic drawing of the reconstructed keratin network in the three cell types investigated.

### Brightness of keratin filament segments

Brightness levels correspond to filaments thickness and are therefore a measure of keratin filament bundling. The observed variation in filament bundling in the different cell types may therefore be a consequence of differences in the local force balance. This force balance is the result of intrinsic cell type-specific properties such as cell shape and arrangement of the actomyosin system and is impacted by forces imposed by neighboring cells and the extracellular matrix. This is supported by the observed differences in bundling of the apical, basal and perinuclear keratin filament networks. However, the variation in filament thickness may also reflect other properties such as the propensity of the network to reorganize, for example, at different times of the cell cycle. Correlating the derived image data with information on mechanical keratin filament properties obtained by laser cutting experiments, force-sensor measurements or traction force analyses may help to better understand the relationship between filament bundling and mechanics.

### Length of keratin filament segments

Keratin filament segment length is an indirect measure of mesh size and may therefore describe another crucial mechanical property of the network. We were surprised to find that the segment length varied only by 10% between the different cell types and thus showed less variation than filament brightness or curvature. This means that the mesh size is fixed within a relatively small range, although the number of meshes per volume differed significantly between the different cell types. A conclusion of this observation is that mesh size may be determined by intrinsic keratin filament properties such as elastic modulus or branching capacity. However time-lapse fluorescence analyses (e.g. *Wöll et al., 2005*) have shown that small meshes can merge and enlarge by the combination of nodes from two segments, forming thicker filament bundles. Conversely, large meshes are subdivided into small meshes by recruitment of segments into a node. A self-regulatory mechanism may thus control the balance between enlargement and reduction of individual meshes.

### Branching of keratin filament segments

Experimental and mechanistic evidence has been provided for keratin filament branching (*Beil et al., 2005*; *Feng and Coulombe, 2015*; *Lee et al., 2012*; *Martin et al., 2016*; *Nafeey et al., 2016*). The fact that mostly two types of branching/intersections were detected in this study, points to a limited repertoire of connectivity. Furthermore, the Y- and X-shaped configurations may be fundamentally different. The Y-shaped joining of three segments may be linked to the recently proposed bundling mechanism involving electrostatic and hydrophobic interactions (*Haimov et al., 2020*). On the other hand, the X-shaped intersections may be related to the disulfide-bond-mediated interactions reported for keratins 5/14 (*Lee et al., 2012*). In this instance, preventing these links resulted in

altered keratin filament network dynamics and epidermal tissue homeostasis (*Feng and Coulombe, 2015*; *Guo et al., 2020*).

## Curvature of keratin filament segments

Bending of keratin filaments provides important information on whether keratins are stretched, relaxed or compressed. Keratin intermediate filaments are highly elastic (*Kreplak et al., 2005*; *Wagner et al., 2007*; *Yamada et al., 2003*). They can be extended under tension forming straight filaments and buckle under compression ( = negative tension) (*Fudge et al., 2008*). The measurements of segment curvature can therefore be taken as indicators of local keratin filament tension. Straight segments were almost completely absent in all cells suggesting overall negative tension of the keratin networks. The observation that filament bundle thickness and curvature are not correlated suggests that the 10 nm filaments are only loosely connected in keratin bundles. The degree of filament bending, however, differed between the different cell types indicating cell type-specific differences in intrinsic force balance and may suggest that the different networks retain different degrees of flexibility before they become stretched to function as mechanical counterbalances to strain. Curvature of keratin filaments also allowed to estimate the persistence length of network-embedded keratin filaments, that is the minimum length between two points on the polymer where they become independent of one another. The value of ~2 μm was only slightly higher than what was reported for other intermediate filaments including keratin filaments (*Block et al., 2015*; *Wagner et al., 2007*). This is low in comparison to actin filaments (~10 μm) (*De La Cruz and Gardel, 2015*; *Isambert et al., 1995*; *Takatsuki et al., 2014*) and microtubules (~5000 μm) (*Hawkins et al., 2010*). This result emphasizes the maintained high flexibility of keratin filaments despite their connectivity through nodes.

## Orientation of keratin filaments

To understand the response of the keratin network to internal and external forces, it is important to have measures of filament orientation. We suggest that uniformly radial orientation of keratin filament segments indicates that cells experience mechanical forces that are homogenous in all directions. In contrast, a preferential orientation in one direction indicates that keratin filaments align in response to mechanical cues that may be induced by unevenly distributed cell-cell contacts, polarized cell-extracellular matrix adhesions or contractile activity. Our data therefore indicate that HaCaT cells are subjected to mechanical strain (uneven azimuth distribution and large sum vector), whereas MDCK and RPE are subjected to little mechanical strain (uniform azimuth distribution and small sum vector). Fine mapping of the distribution patterns of hemidesmosomes and desmosomes may help to understand the differences and to initiate mechanical modelling, improving previous models of keratin network, morphogenesis, dynamics and mechanics (*Beil et al., 2009*; *Dallon et al., 2019*; *Latorre et al., 2018*; *Portet, 2013*; *Portet et al., 2019*; *Portet et al., 2015*).

The observed high degree of segment property variability must be taken as an indication of the amazing plasticity of the keratin filament system. The node/segment arrangement revealed a unique 3D structure that can be modified in multiple ways affecting bundling, flexibility, and local cross-talk. Keratin isotypes, posttranslational modifications, external and internal forces, and interactions with other cytoskeletal and cellular components may determine these properties and thereby act as modulators of network structure at different length and time scales.

## Materials and methods

### Cell culture

Madin-Darby Canine Kidney (MDCK) cells were stably transfected with a plasmid encoding human full-length keratin eight with an EYFP tag at its carboxyterminus (*Windoffer et al., 2004*; recloned into pEYFP-N1 (Clontech) with BamHI and EcoRI. MDCK cells were obtaned from DKFZ (Deutsches Krebsforschungszentrum). Resulting cell clone H9 was used for the analyses described in this paper. Cells were grown in Dulbecco's Modified Eagle Medium (DMEM, Sigma-Aldrich) with 10% (v/v) fetal calf serum (Pan Biotech) including 700 μg/ml G418 (Sigma-Aldrich) for selection. Vital MDCK cells were imaged after 3 days when the monolayer had reached complete confluence.

Immortalized human HaCaT keratinocyte cell clone B10 expressing EYFP-tagged human keratin 5 (HK5-EYFP) has been described (*Moch et al., 2013*). HaCaT cells were obtained from the Fusenig lab, which first isolated and described this cell line (*Boukamp et al., 1988*). The cells were grown in DMEM containing l-alanyl-glutamine (Sigma-Aldrich) supplemented with 10% (v/v) fetal calf serum (SeraPlus; PAN Biotech) and passaged as described (*Moch et al., 2013*). Vital HaCaT cells were imaged 1–2 days after reaching confluence corresponding to 5–7 after seeding. Both cell lines were tested by PCR to be mycoplasma free.

## Preparation of retinal pigment epithelium

Retinas were prepared from homozygous keratin 8-YFP knock-in mice, which synthesize fluorescence-tagged keratin eight from the modified endogenous gene locus (*Schwarz et al., 2015*). Mice were sacrificed by cervical dislocation and eyes were enucleated using curved forceps. Eyeballs were then fixed with 2% paraformaldehyde in rotation at 4 °C for 4 hr. For retina flat-mount preparation, the eye balls were dissected as previously described (*Claybon and Bishop, 2011*). Briefly, cornea and lens were removed and the eyecup was cut into a multileaved flower shape. The neural retina was removed and the remaining tissue was mounted on a microscope slide with Mowiol (Sigma-Aldrich).

## Image recording

To obtain statistically relevant data, ten image data sets were recorded of each cell type with a Zeiss LSM710 using airyscan mode (Carl Zeiss, Jena, Germany). An oil immersion Plan-Apochromat 63 x/1.4-NA, DIC M27 objective (Carl Zeiss) was used. Fluorescence was detected with an 488 nm argon laser (module LGK 7872 ML8) and filter set BP 495–550 using 2.5% power. Optimal factory settings for z-stacks were used resulting in a voxel size of xy = 0.044–0.065 µm and z = 0.17–0.2 µm. No further image optimization was used. To assure that only single cells were analyzed, the fluorescence in individual cells was cropped from confluent MDCK monolayers and retinal pigment epithelium (RPE) using Fiji standard tools. Cell borders were manually defined for each z-slice and the cell exterior was set to a grey value of 0. Wild-type HaCaT and HK5-EYFP producing HaCaT clone B10 cells were mixed prior to imaging to prevent overlap of fluorescence signals in neighbouring cells.

## Segmentation of image stacks

TSOAX software (*Xu et al., 2019*; *Xu et al., 2015*; *Xu et al., 2011*) was used for the segmentation of filaments from the recorded 3D image stacks (*Figure 1—figure supplement 2A*). TSOAX facilitates quantitative analysis of network dynamics of multi-dimensional biopolymer networks imaged by various microscopic imaging modalities. The underlying methods of TSOAX include multiple Stretching Open Active Contour (SOAC) models for extraction and a combined local and global graph-matching framework (*Li et al., 2010*). For optimal results, the default TSOAX parameters were used, with the exception of 'external-factor' which was set to two based on internal testing. Since computation time can be extremely long, TSOAX was executed in parallel instances on the supercomputer CLAIX at RWTH Compute Cluster (https://www.itc.rwth-aachen.de/go/id/eucm).

## Calculation of a segment/node-based network model

To get a perfect network representation, it is necessary to divide the network into defined pieces ( = segments) that do not cross each other, and can be connected at their ends by nodes. Segmentations using TSOAX defines polygonal chains with *n* vertices ( = snakes). Each vertex is defined in x, y, z and has an associated density ( = fluorescence intensity). However, the snakes are not associated with each other via nodes and may cross each other. (*Figure 1—figure supplement 2B*). To obtain a comprehensive and coherent numerical network definition from snakes, KerNet software was developed. It generates a graph representation consisting of segments (~ edges) and nodes. To compute such a connected network representation, vertices of a snake that lie within $\varepsilon$ = 1.1 x voxelsize distance of one another are combined to form a node. If more than two vertices are within ε distance, a candidate set is created. Any vertex, which is at most ε away from a vertex in the candidate set, is added to the set until this is no longer possible. Then all vertices in the candidate set are combined to form a single node. Finally, any start or end vertex of a segment is a node. If two nodes are connected by a part of a snake then a segment connects the two nodes in the network. In contrast to classical edges, which are defined as lines connecting the nodes, each segment is a list of vertices (x,y,z) and their

associated densities. The segments therefore inform about the trajectory of the snake between its two endpoints and the density progression along the segment. To summarize, nodes represent the first and last vertex of a segment and mark the connection between segments whose endpoints are close (*Figure 1—figure supplement 2C*). All segments and nodes together represent a Euclidean 3D network map of a given cell.

## Validation of segmentation

For validation of the segmentation, the original fluorescence image stacks were compared with the segmented 3D data. All vertices retrieved from KerNet were plotted into the original image stacks. The z-values of the vertices were rounded to the nearest z-slice and all segment vertices were drawn as discs with a diameter proportional to their brightness. To simulate the blurriness of the raw images, the vertex stacks were 3D-Gauss filtered using Fiji (values 5,5,3).

## Cinematic rendering of segments

For inspection of the 3D network representation, the xyz-position data of segments were exported into OBJ files (http://www.martinreddy.net/gfx/3d/OBJ.spec), which, in turn, can be imported into 3D rendering programs. For the representation of segment brightness as renderable tubes, density properties of segments were compiled into 3ds Max (Autodesk) script files that translate segment brightness into tube thickness, resulting in renderable splines for each segment. In some instances, random colors were assigned to each segment (*Figure 1—figure supplement 2D*). Finally, 3ds Max rendering was used for the generation of stills and videos.

## Virtual reality

For further visual inspection of the keratin networks, a basic Virtual Reality (VR) application was designed. The VR application was implemented using Unreal Engine 4 (https://www.unrealengine.com/en-US/). It displays the datasets of cells as 3D models. The individual filaments are visualized as tubes. Their brightness is encoded in the tubes' thickness. The resulting structure of interconnected tubes can then be visually inspected in VR, allowing a detailed view on the digitalized subcellular network in any magnification. Individual filaments can be interactively assigned to one of several distinct classes, pre-defined by the domain expert. Color-coding is then used to visually differentiate the classes in the immersive visualization.

During our research, we used the HTC Vive Pro headset to manually classify the filaments of a cell. Afterwards, the aixCAVE at RWTH Aachen University was used to collaboratively discuss and adjust the classification.

## Numerical analysis and visualization

Matlab routines were used for further analysis of network properties. Up to 150,000 vertices may be present per cell. Segment length, distance between nodes, bending, curvature and apparent persistence length of segments were calculated. Bending was defined as the ratio beween the shortest distance between the segment endpoints and the actual segment length. The curvature was calculated as the 2nd derivative of the smoothed tangent vector with respect to the arc length (*Sternberg, 2012*). Since persistence length can only be calculated in free moving segments and the segments are restricted in their movement by the nodes in our situation, we calculated an apparent persistence length as described (*Doi and Edwards, 1990*).

Statistical analyses were performed with Matlab and GraphPad Prism software. For non-Gaussian distribution, Mann-Whitney test was used. Values are expressed as the mean ± SD. * represents a p-value of $p < 0.05$, ** $p < 0.01$, *** $p < 0.001$ and **** $p < 0.0001$. The differences were considered significant when $p < 0.05$. Non-significant differences were indicated by n.s.

To visualize segment properties such as length or curvature, the respective values were plotted and color-coded at the vertex location resulting in heat maps showing the distribution of the selected property. Here, Fiji scripts were used to plot color-coded properties into xyz coordinates of vertices.

For calculation of cell size, a threshold was used, that separated the background fluorescence inside and outside of cells.

## Directional vectors

The directionality of segments was calculated for the keratin network of single cells in two ways:

1. Since the start and end points of segments are interchangeable, segments with an orientation difference of 180 degrees are equally orientated. Normalized histograms were plotted for azimuth and elevation from 0 to 180 degrees. Next, we calculated the deviation of the histogram from a histogram with an even distribution of orientations where each degree has the same frequency. Finally, the difference between the segment histogram and the evenly distributed histogram was calculated as the sum of the differences in each bin.
2. Keeping the segments in their original orientation we analyzed the position of segments in relation to the cell center. The cell center was defined as the mean of the mean positions of all segments in xyz. It was then determined whether the start point or end point of each segment was closer to the cell center. The segment was then moved to the cell center such that the closer point was placed to the cell center. In this way, all segments originate at the cell center and direct outwards. The resulting radial histograms were plotted to depict the distribution of segments in xy and yz directions. Treating each segment as a vector extending from the cell center to the segment's end, the sum vector of all segments was calculated and plotted.

## Software, original and processed Data

Software, original and processed data are available at http://kernet.rwth-aachen.de/ and https://github.com/VRGroupRWTH/Zytoskelett, (copy archived at swh:1:rev:2bb3a72647d36a679d-496907dce61ae0f7c4a544, *Oehrl, 2021*).

## Acknowledgements

We gratefully acknowledge the helpful discussions and advice of Drs. Michael Schaub (RWTH Aachen University), Charlotte Lorenz and Sarah Köster (Göttingen University), and Rudolf Merkel (FZ Jülich). We also thank Michael Anhuth and Till Petersen-Krauß (RWTH Aachen University) for the implementation of the VR application and Ursula Wilhelm (MOCA) for providing the immunoblot. The work was supported by the German Research Council (LE566/18-2; WI173/8-2; GRK2415/363055819). Calculations were performed with computing resources granted by RWTH Aachen University under project rwth0452. J.D.R and T.P. were supported by a grant from the Interdisciplinary Centre for Clinical Research within the faculty of Medicine at RWTH Aachen University.

## Additional information

### Funding

| Funder | Grant reference number | Author |
| --- | --- | --- |
| Deutsche Forschungsgemeinschaft | WI173/8-2 | Reinhard Windoffer |
| Deutsche Forschungsgemeinschaft | LE566/18-2 | Rudolf Leube |
| Deutsche Forschungsgemeinschaft | GRK2415/363055819 | Reinhard Windoffer Nicole Schwarz Sungjun Yoon Teodora Piskova Rudolf Leube |
| RWTH Aachen University | rwth0452 | Reinhard Windoffer |
| Medizinische Fakultät, RWTH Aachen University | IZKF | Teodora Piskova Jacopo Di Russo |

The funders had no role in study design, data collection and interpretation, or the decision to submit the work for publication.

### Author contributions

Reinhard Windoffer, Conceptualization, Data curation, Formal analysis, Funding acquisition, Investigation, Methodology, Project administration, Resources, Software, Supervision, Validation, Visualization, Writing – original draft, Writing – review and editing; Nicole Schwarz, Sungjun Yoon,

Teodora Piskova, Investigation; Michael Scholkemper, Johannes Stegmaier, Software; Andrea Bönsch, Software, Visualization; Jacopo Di Russo, Conceptualization; Rudolf E Leube, Conceptualization, Funding acquisition, Project administration, Writing – original draft, Writing – review and editing

### Author ORCIDs
Reinhard Windoffer ⬥ http://orcid.org/0000-0003-1403-5880
Sungjun Yoon ⬥ http://orcid.org/0000-0001-7363-3261
Teodora Piskova ⬥ http://orcid.org/0000-0002-5143-2634
Michael Scholkemper ⬥ http://orcid.org/0000-0002-3669-8119
Johannes Stegmaier ⬥ http://orcid.org/0000-0003-4072-3759
Andrea Bönsch ⬥ http://orcid.org/0000-0001-5077-3675
Jacopo Di Russo ⬥ http://orcid.org/0000-0001-6731-9612
Rudolf E Leube ⬥ http://orcid.org/0000-0002-5519-7379

### Ethics

All animal experiments were conducted in accordance with the guidelines for the care and use of laboratory animals and were approved by the Landesamt für Natur, Umwelt und Verbraucherschutz Nordrhein-Westfalen (LANUV; reference number 84-02.04.2015.A190 and approvals according to §4 of the German Animal Welfare Act).

### Decision letter and Author response

Decision letter https://doi.org/10.7554/eLife.75894.sa1
Author response https://doi.org/10.7554/eLife.75894.sa2

## Additional files

### Supplementary files
• Transparent reporting form

### Data availability

Software, original and processed data are available at http://kernet.rwth-aachen.de/ and https://github.com/VRGroupRWTH/Zytoskelett (copy archived at swh:1:rev:2bb3a72647d36a679d496907dce61ae0f7c4a544).

The following dataset was generated:

| Author(s) | Year | Dataset title | Dataset URL | Database and Identifier |
| --- | --- | --- | --- | --- |
| Windoffer R | 2021 | KerNet, 3D fluorescence and segmentation datasets of keratin networks from MDCK, HaCaT, and RPE cells | https://dx.doi.org/10.5061/dryad.3xsj3txht | Dryad Digital Repository, 10.5061/dryad.3xsj3txht |

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
