## [Editor Report]

In this body of work you have skillfully adapted and developed tools for the three-dimensional visualization and systematic analysis of the entire keratin filament network in three different types of cells. The resulting contribution is original, provides insight at both a methodological and biological level, and nicely complements and extends emerging information about the high resolution structure of intermediate filaments in situ (by cryoelectron tomography / Medalia lab, Switzerland). The manuscript is well-written, well-illustrated, and the authors are thorough in their recognition of previous studies of relevance to their own. We believe that this article will be foundational in the specialized field of intermediate filament biology and will have a significant impact in the broad field of cell biology.

---

## [Decision Letter]

**Decision letter after peer review:**

Thank you for submitting your article "Quantitative Mapping of Keratin Networks in 3D" for consideration by *eLife*. Your article has been reviewed by 3 peer reviewers, one of whom is a member of our Board of Reviewing Editors and the evaluation has been overseen by Anna Akhmanova as the Senior Editor. The following individual involved in the review of your submission has agreed to reveal their identity: John E Eriksson/John (Reviewer #2).

Essential revisions:

1) Congratulations to the authors on a beautiful and important body of work.

2) The authors are expected to address all of the comments that were deemed "required" by the three reviewers, each of whom is an expert in the field and provided a balanced and informed opinion about the work.

3) The issue of whether to retain Figure 3 in the main space of the paper (cf. reviewer #3) or relocate it to supplementary material is one worth thinking about. As the reviewing editor I support reviewer 3's opinion that Figure 3 does not belong to the main space of the paper, though I would like to see it accessible to the public. The editor is expected to weigh in on this specific issue.

4) There are a number of additional comments and suggestions that were provided by the three reviewers. The authors should make a deliberate effort to address them. This contribution has the clear potential of becoming a landmark paper in the field and addressing (many of) these comments would help make the contribution clearer, stronger, and potentially more impactful.

*Reviewer #1 (Recommendations for the authors):*

Issues that should be addressed by the authors include:

Resolution of the microscopic imaging. Can the authors detect single 10 nm filaments in the cells they studied? If not, is there a sense of how many filaments are comprised in the segments they can detect, microscopically?

Can the author convert their whole cell imaging data into a concentration of keratin proteins, and number of filaments, inside cells? Some assumptions would need to be made, but it would be interesting to compare the outcome of this analysis to biochemical measurements that have been published, e.g., for basal keratinocytes in the epidermis in situ (see X Feng et al., J Invest Dermatol 2013 PMID 23190880). How does the imaging-derived information compare to actual (biochemical) keratin concentrations in the cell lines analyzed (MDCK, HaCaT)? This comparative analysis would provide a sense of "how much of the keratin pool" is being captured through the imaging approach developed by the authors.

Section 3.6. Is there any trend regarding subcellular distribution for the Y-shape, X-shape, star-shaped and more complex arrangements of intersections between filaments?

Section 3.7. One would expect that thicker segments show a lower degree of bending, as they are likely to represent bundles of filaments. Comments? A key determinant, as pointed out by the authors in their Discussion, is whether fiber segments are subjected to forces (e.g., axial compression, lateral pulling, etc.).

Bundling properties. Given the known arrangement of subunits in the keratin filament lattice, there likely is a fair amount of GFP moieties in the filaments analyzed. Does the presence of a GFP fusion, which presumably is exposed at the surface of filaments, impact their ability to undergo bundling? (of relevance: what is the stoichiometry of fusion protein relative to endogenous K8 and other type II keratins?).

Line 136 – Imaging of HaCaT cells. "When" were the analyses performed relative to time in culture and degree of confluence?

Line 294. What do the authors mean by "good correlation"?

Segment 3.8. Orientation. The authors may wish to illustrate the terminology they use, e.g., in a schematic presented in the setting of a suppl. figure.

Discussion – line 446 – one wonders why the authors did not routinely mix fluorescently labeled and non-labeled cells to avoid the edge effects they describe.

Line 456. Please define "motion parallax".

*Reviewer #2 (Recommendations for the authors):*

– Can you give an assumption why more segments can be detected in HaCaT cells? Why the average length was shortest in MDCK cells and longest in RPE cells? How does it link to the biological feature or function of the cells? if we have a random cell given, can you infer the biological property of this cell?

– It is stated in Abstract and Introduction that super-resolution imaging was used for image capture, but the Materials and methods section states that Airyscan was used, which is not a real super-resolution technique. Later on "high resolution images from Airyscan" is used, which would be more accurate

– In the "2.10 Directional vectors" section, the authors describe how the directionality of the segments was calculated. This section does not come across as detailed enough as it seemed challenging to decipher how this works out. Grateful for some added details.

– In some of the renderings in the images (especially clear in Figure 4), it possible to see that it has not formed lines, but dots in a line. Any reason for this? Only partially successful segmentation or something else?

– Interesting to know if one could observe any differences between keratin in physiological state and diseased state using this type of modelling, for example, non-cancerous cells vs. cancer cells, or cells in confluent state vs. a state mimicking wound healing?

*Reviewer #3 (Recommendations for the authors):*

1. After a careful inspection of Figure 1A, it seems that the position of the vertices are not always at the center of the fluorescence signal, shouldn't it be at the very center of the original signal?

2. The fluorescent signal is translated into a tube, however:

a. In your images, similar intensities have sometimes different dimensions (thicker thinner), this is not always reflected in the thickness of the rendered structure.

b. In Y-shape junction, shouldn't one bundle should be thicker?

c. Why is it correct to estimate bundles as tubes and not with oval shapes?

d. Since the signal of an individual keratin filament can be detected, why the thickness of the rendered tube is proportional to an (at least) estimated number of filaments? even a naïve integer multiplication of the signal would give a more realistic view of the network.

3. This reviewer does not see any scientific incentive to including figure 3.

4. The curvatures of the segments are slightly different between the cell types studied. It is not clear what would that means or how would it influence the keratin network?

5. The authors indicate the total length of the keratin bundle network in cells, but a reference number is missing. How is it in comparison to actin and microtubules?

6. Are there any significant variations between the keratin network, e.g. density or segment directionality, at the vicinity of the nucleus in comparison to other cytoplasmic regions?

---

## [Author Response]

Essential revisions:1) Congratulations to the authors on a beautiful and important body of work.2) The authors are expected to address all of the comments that were deemed "required" by the three reviewers, each of whom is an expert in the field and provided a balanced and informed opinion about the work.3) The issue of whether to retain Figure 3 in the main space of the paper (cf. reviewer #3) or relocate it to supplementary material is one worth thinking about. As the reviewing editor I support reviewer 3's opinion that Figure 3 does not belong to the main space of the paper, though I would like to see it accessible to the public. The editor is expected to weigh in on this specific issue.

As suggested, we moved Figure 3 to the Supplementary Material section.

4) There are a number of additional comments and suggestions that were provided by the three reviewers. The authors should make a deliberate effort to address them. This contribution has the clear potential of becoming a landmark paper in the field and addressing (many of) these comments would help make the contribution clearer, stronger, and potentially more impactful.Reviewer #1 (Recommendations for the authors):Issues that should be addressed by the authors include:Resolution of the microscopic imaging. Can the authors detect single 10 nm filaments in the cells they studied? If not, is there a sense of how many filaments are comprised in the segments they can detect, microscopically?

We can detect single 10 nm filaments by fluorescence microscopy but are unable to determine the number of filaments within a given keratin bundle. Quantification of filament bundle brightness did not reveal a stepwise but a continuous distribution.

Can the author convert their whole cell imaging data into a concentration of keratin proteins, and number of filaments, inside cells? Some assumptions would need to be made, but it would be interesting to compare the outcome of this analysis to biochemical measurements that have been published, e.g., for basal keratinocytes in the epidermis in situ (see X Feng et al., J Invest Dermatol 2013 PMID 23190880).

We thank the Reviewer for helping us to connect our results with previous observations. We added the following section:

“3.6 Estimation of cellular keratin contentIt is now possible to estimate the amount of keratin from our observations and to compare it to the previously estimated 8.07 ± 1.88 pg keratin in single basal keratinocytes of the murine epidermis (Feng et al., 2013). […] Performing the same type of calculation we deduced a keratin content of 1.3 pg or 5.0 pg per MDCK cell and 1.2 or 5.5 pg for single RPE cells.”How does the imaging-derived information compare to actual (biochemical) keratin concentrations in the cell lines analyzed (MDCK, HaCaT)? This comparative analysis would provide a sense of "how much of the keratin pool" is being captured through the imaging approach developed by the authors.We appended an immunoblot in new SupplementaryFigure 1—figure supplement 1 showing the relative amount of endogenous keratin 8 in the newly prepared MDCK cell line and added the following statement in section 2.1:“Immunoblot analysis of total cell lysates revealed comparable amounts of YFP-tagged and endogenous keratin 8 (Figure 1—figure supplement 1).”

We furthermore emphasize in section 2.1 that the entire keratin 8 is YFP-tagged in RPE cells:

“To examine the 3D organization of the keratin intermediate filament network in its native tissue context, we made use of the recently described knock-in mouse line expressing only YFP-tagged but no wild-type keratin 8.”

Section 3.6. Is there any trend regarding subcellular distribution for the Y-shape, X-shape, star-shaped and more complex arrangements of intersections between filaments?

We could not find a preference of subcellular distribution for the different shapes.

Section 3.7. One would expect that thicker segments show a lower degree of bending, as they are likely to represent bundles of filaments. Comments? A key determinant, as pointed out by the authors in their Discussion, is whether fiber segments are subjected to forces (e.g., axial compression, lateral pulling, etc.).

We did not find a correlation between keratin bundle thickness and bending and therefore added the following statements:

- Previous section 2.7/new section 2.8:

“We did not see a correlation between filament bundle thickness and curvature (pairwise Pearson correlation coefficient in MDCK = -0.10, HaCaT = 0.08, RPE = 0.14).”

- Result section:

“The observation that filament bundle thickness and curvature are not correlated suggests that the 10 nm filaments are only loosely connected in keratin bundles.”

Bundling properties. Given the known arrangement of subunits in the keratin filament lattice, there likely is a fair amount of GFP moieties in the filaments analyzed. Does the presence of a GFP fusion, which presumably is exposed at the surface of filaments, impact their ability to undergo bundling? (of relevance: what is the stoichiometry of fusion protein relative to endogenous K8 and other type II keratins?).

As specified in our answer above, the YFP-tagged keratin 8 makes up to 100% of keratin 8. While we cannot formally exclude a mild effect on bundling, the current manuscript together with many publications not only from our own lab but also many other labs world-wide indicate that fluorescent protein-tagged keratins are highly reliable, precise and fully functional reporters of the wild-type polypeptides. The healthy and inconspicuous homozygous Krt8-YFP mouse colony further attests this conclusion.

For the Reviewers’ perusal we provide a comparison of an anti-keratin immunostaining of MDCK cells (left) with one of the MDCK cells analysed in the current manuscript (right). While the quality of the immunodetection is inferior to that of the fluorescence in the viable cell because of the necessary fixation and permeabilisation, overall network morphology and filament bundling appear to be comparable.

**Author response image 1. sa2fig1:** 

Line 136 – Imaging of HaCaT cells. "When" were the analyses performed relative to time in culture and degree of confluence?

HaCaT B10 cells were imaged 1-2 days after reaching confluence corresponding to 5-7 after seeding. This information is now included in section 4.1.

Line 294. What do the authors mean by "good correlation"?

We now write “…that the recorded and derived numerical 3D representations of the keratin filament network are in agreement.”.

Segment 3.8. Orientation. The authors may wish to illustrate the terminology they use, e.g., in a schematic presented in the setting of a suppl. figure.

We provide a scheme in new Supplementary Figure 7 suppl 1 illustrating the terminology used in the dissection of the complex 3D keratin network orientation.

Discussion – line 446 – one wonders why the authors did not routinely mix fluorescently labeled and non-labeled cells to avoid the edge effects they describe.

While it is obviously not possible to do mixing in the native RPE tissue context, we failed to mix cells in our initial analyses of MDCK cells. Obviously, mixing would have been advantageous.

Line 456. Please define "motion parallax".

We now write:

“… (iii) head tracking provides enhanced motion parallax. Motion parallax is a monocular depth cue which allows observers to perceive the depth of the visualized cell from their own motion. The filaments close to the observers are perceived to be moving faster than filaments in the distance.”

Reviewer #2 (Recommendations for the authors):– Can you give an assumption why more segments can be detected in HaCaT cells? Why the average length was shortest in MDCK cells and longest in RPE cells? How does it link to the biological feature or function of the cells? if we have a random cell given, can you infer the biological property of this cell?

These are clearly the most important but also most difficult questions. Correlation of network architecture with functional readouts such as cellular mechanics, signalling, proliferation, differentiation, stress and others are needed to address these questions. Our tools allow to investigate such aspects and serve as a strong motivation for our ongoing research.

– It is stated in Abstract and Introduction that super-resolution imaging was used for image capture, but the Materials and methods section states that Airyscan was used, which is not a real super-resolution technique. Later on "high resolution images from Airyscan" is used, which would be more accurate

In spite of the advertisement by Zeiss (https://www.zeiss.com/microscopy/int/products/confocal-microscopes/lsm-980.html), we substituted “super-resolution” by “high resolution” in the context of Airyscan technology.

– In the "2.10 Directional vectors" section, the authors describe how the directionality of the segments was calculated. This section does not come across as detailed enough as it seemed challenging to decipher how this works out. Grateful for some added details.

As specified in our response to Reviewer 1, we added another scheme to illustrate aspects of the vector orientation in new Supplementary Figure (Figure 7—figure supplement 1).

– In some of the renderings in the images (especially clear in Figure 4), it possible to see that it has not formed lines, but dots in a line. Any reason for this? Only partially successful segmentation or something else?

In the 3D renderings vertices are connected as tubes (e.g., Figure 2), whereas the heatmaps in Figures 3, 4, 6 and 7 depict properties of individual vertices that are not evenly spaced and are therefore represented as puncta in certain regions.

– Interesting to know if one could observe any differences between keratin in physiological state and diseased state using this type of modelling, for example, non-cancerous cells vs. cancer cells, or cells in confluent state vs. a state mimicking wound healing?

We wholeheartedly agree with the Reviewer that these are pressing issues to be answered but would like to emphasize that reliable evaluation requires careful and detailed analyses.

Reviewer #3 (Recommendations for the authors):1. After a careful inspection of Figure 1A, it seems that the position of the vertices are not always at the center of the fluorescence signal, shouldn't it be at the very center of the original signal?

The Reviewer correctly points out that there is local imprecision in the co-localization between the segmentation data and original fluorescence data. This is due to the transformation of the slice-based fluorescence z-positions into continuous positions in the segmentation data. Thus, ‘new’ z-positions are created in between the slices of the original z-stack positions in the segmentation data leading to slight incongruences in the overlays of the projection images.

2. The fluorescent signal is translated into a tube, however:a. In your images, similar intensities have sometimes different dimensions (thicker thinner), this is not always reflected in the thickness of the rendered structure.

We do not quite understand the point of criticism. Fluorescence brightness was taken as a measure of filament thickness. This was done consistently without any manual interference. To further clarify this we added the following statement in section 2.4:

“We will therefore use the terms brightness and thickness interchangeably.”

b. In Y-shape junction, shouldn't one bundle should be thicker?

Unfortunately, the data do not support this obvious concept.

c. Why is it correct to estimate bundles as tubes and not with oval shapes?

The choice for a circular geometry was based on limited data (e.g., Hémonnot, et al., *ACS 2016, Nano* 10,: 3553–61. https://doi.org/10.1021/acsnano.5b07871) and appears to be the most simple way to represent filament bundles. It is fully consistent with the lack of asymmetry in our data sets. We are also not aware that other geometries such as an oval or square arrangement have been discussed in the literature.

d. Since the signal of an individual keratin filament can be detected, why the thickness of the rendered tube is proportional to an (at least) estimated number of filaments? even a naïve integer multiplication of the signal would give a more realistic view of the network.

We would like to point out that the microscopic resolution was insufficient to resolve filament thickness. Furthermore, as detailed in our answer to Reviewer 1, we did not find a stepwise distribution of filament brightness.

3. This reviewer does not see any scientific incentive to including figure 3.

We moved Figure 3 into the supplements.

4. The curvatures of the segments are slightly different between the cell types studied. It is not clear what would that means or how would it influence the keratin network?

As stated in the Discussion, we interpret differences in curvature as indicators of positive and negative tension on keratin filaments.

5. The authors indicate the total length of the keratin bundle network in cells, but a reference number is missing. How is it in comparison to actin and microtubules?

To our knowledge there are no solid data on the total length of actin filaments and microtubules in single cells.

6. Are there any significant variations between the keratin network, e.g. density or segment directionality, at the vicinity of the nucleus in comparison to other cytoplasmic regions?

We hereby repeat our response to Reviewer 1:

Exploration of subdomains is afforded by the interactive 3D renderings provided at KerNet.rwth-aachen.de. However, systematic and quantitative analyses of segment properties in subcellular domains is an obvious but quite challenging issue. The main difficulty is a precise spatial definition of subdomains in 3D, which would require substantial effort.